# CSR Disclosures, CSR Awards and Corporate Governance as Determinants of the Cost of Debt: Evidence from Malaysia

**Shyamala Dhoraisingam Samuel [1,*]** , **Sakthi Mahenthiran [2]** and **Ravindran Ramasamy [3]**

1   School of Business, Monash University Malaysia, Subang Jaya 47500, Malaysia
2   College of Business Administration, Gulf University for Science and Technology, Mubarak Al-Abdullah Area, West Mishref 40006, Kuwait
3   Graduate Business School of Business, University Tun Abdul Razak (UNIRAZAK), Kuala Lumpur 50480, Malaysia
*   Correspondence: shymala.dhoraisingam@monash.edu

**Abstract:** The current study examines the relationship between corporate social responsibility (CSR) disclosures, media announcements of CSR awards and a firm's cost of debt in Malaysia. The sample consists of 104 Malaysian publicly listed companies belonging to the Edge Billion Ringgit Club between 2009 to 2015. The study uses panel data regression analysis and the ordinary least squares estimation method. The results find that the interaction between CSR disclosures and winning awards for a company's CSR initiatives and practices lowers the cost of debt. Our study concludes that, when a company discloses more information on its CSR initiatives and practices, it reduces the cost of debt. Thus, we argue that CSR disclosures and awards can act as proxies for branding listed firms, making them more marketable and allowing them to negotiate better debt contracts. However, the study shows that politically connected companies have a higher debt cost than non-politically connected firms. Furthermore, our results indicate that smaller boards are effective, but audit committees are not effective in monitoring the board of directors in Malaysian listed firms.

**Keywords:** corporate governance; corporate social responsibility; reputation; CSR awards; cost of debt; Malaysia

## 1. Introduction

Efforts to transform Malaysia from being based on agricultural to being based on an export-led economy has led to the initiation of national development strategies such as the New Economic Policy (NEP, 1971–1990), National Development Policy (NDP, 1991–2000), National Vision Policy (NVP, 2001–2010) and the New Economic Model (NEM), introduced in 2009. These national policies enabled Malaysia to attain a high income and the developed nation status by 2020. This study covers a period prior to 2020, when the government's role in introducing these economic reforms raised public concerns about sustainability. For example, under the 10th Malaysian Plan, the oil and gas sector was identified as 1 of the 12 key sectors contributing to economic growth. As a result, oil and gas companies exploited the production of fossil fuels to meet national aspirations, which, at the same time, caused externalities to society, such as pollution and global warming (Oh et al. 2010)[1]. These externalities have caused by business operations to conflict with the National Outline Perspective Plan (2001–2010), which states that Malaysian Public Listed Companies (PLCs) must balance their economic growth with sustainable social and environmental considerations. The 10th Malaysian Plan (2011–2015) encourages Government-Linked Companies (GLCs) and the private sector to actively steer economic growth and sustainable development to aspire to achieve Vision 2020. This study investigates how PLCs' corporate social responsibility (CSR) initiatives and how their disclosures affect the cost of debt capital and the role that banks and other institutions play in helping PLCs balance their growth with CSR considerations.

The allocation of scarce resources can be perceived from two perspectives. The positive perspective is that companies engaging in CSR activities and practices increase their shareholder value, whereas the negative view is that they are seen as costly diversions of scarce company resources (Goss and Roberts 2011; Friedman 1970). For example, in promoting the concept of a *Green Economy*, the Malaysian government established the Green Technology Financing Scheme (GTFS). The fund enables producers and users of green technology to obtain soft loans. Moreover, the government subsidizes 2% of the interest rate and provides a guarantee of 60% on the amount of financing. By engaging in green technology policy initiatives, the benefit a company receives takes the form of seeking legitimacy, improving its reputation and brand image and engaging with its stakeholders (Magnanelli and Izzo 2017). The voluntary adaption of these initiatives under the 9th and 10th Malaysian Plans (MP) may aid companies in achieving a competitive advantage (Greening and Turban 2000), which in turn may lower the cost of borrowing by PLCs. However, to date, no studies have addressed the consequences of allocating scarce resources to align with the government's initiatives and policies in Malaysia.

Clarkson et al. (2013) and Malik (2015) have suggested that the strategic use of CSR activities and practices can maximize firm value. Dhaliwal et al. (2012) noted that CSR activities reduce costs and improve operational efficiency. Hence, when companies invest in the government's national initiatives and policies, they can reduce their operational and financial risks, which can reduce the cost of debt and equity capital. Richardson and Welker (2001) examined the relationship between financial and social disclosures and the cost of equity capital for a sample of Canadian firms. Their results show that CSR disclosures can directly influence the cost of capital either through investor preference for socially responsible and ethical investing or through reduced information asymmetry affecting estimation risk. Padgett and Galan (2010) used resource-based theory to examine the relationship between firm level attributes and the extent of CSR investments. Their findings show that a company's resource availability and profitability determine the extent of its CSR investments. The Malaysian study by Smith et al. (2007) shows that firm profitability increases environmental disclosures.

Similarly, Saleh et al. (2011) showed that Malaysian companies investing in CSR practices improve their financial performance, which, in the long term, builds customer loyalty. The study by Cormier et al. (1993) based in the US, a developed economy, shows that the debt-to-equity ratio or financial leverage significantly impacts environmental disclosure. CSR initiatives enable a firm to establish its corporate identity and reputation to institutionalize its CSR activities, which can help with the legitimization of PLCs (Subramaniam et al. 2016).

Developed countries such as the US, UK and Australia face pressure from the public and stakeholders regarding the level of CSR information disclosed, unlike in a developing country such as Malaysia, where government incentives are important (Ali et al. 2017; Ghazali 2007). One of the reasons attributed to this is that CSR is still in its developing stage in emerging countries (Amran and Susela Devi 2008). Financial and non-financial disclosures provide information on firm value by reducing the cost of capital (Dhaliwal et al. 2012). Furthermore, their results show that CSR (nonfinancial) disclosure plays a complementary role in improving financial transparency. We have extended the research on CSR reporting by examining whether winning CSR awards plays a complementary role in financial transparency and accountability in the Malaysian context, particularly in lowering the cost of debt. CSR awards may credibly authenticate the additional nonfinancial information that aids in assessing PLC marketability and financial performance, thereby also allowing them to negotiate better debt contracts at lower costs. Malaysia provides a good setting to explore this question because the blockholding of shares held by family members, state investments, Government-Linked Companies (GLCs) and institutional ownership may help authenticate CSR awards' credibility. For example, winning CSR awards can be viewed as coercive pressure arising from political interference from the government.

The motivation for the current study arises from the 9th and 10th MP policies and the initiatives relating to sustainability issues. These initiatives introduced by the government

are discretionary to a company. If a company discloses its CSR activities and practices that align with the MPs, the information is helpful to stakeholders' decision making. Users of annual reports were surveyed by Deegan and Rankin (1999) in Australia to explore whether the disclosures of environmental information were useful for decision making. Users sought information on whether to invest or lend funds, to provide further resources such as labor or to continue to support a firm by purchasing its products or services. In addition, stakeholders perceive the operating risks of the company to be lower and, at the same time, allow the company to gain legitimacy within a society (Subramaniam et al. 2016; Ye and Zhang 2011; Lindblom 1994). To encourage the development and growth of the key economic sectors identified in the 10th MP, the funding for these capital projects was from the government via the Economic Planning Unit, and the Ministry of Finance provided initiatives. In addition, the Malaysian Industrial Development Authority (MIDA) was set up to attract foreign investment and to promote domestic investment that aids national development and growth.

The second motivation for this study was that there is no formal rating system in Malaysia, unlike in the U.S., where rating agencies such as Kinder–Lydenberg–Domini (KLD) exist to authenticate the credibility of CSR disclosures. Hence, winning awards may be perceived by society as a benchmark for the "Best Practices" of Malaysian PLCs on the Main and Second Boards of the Kuala Lumpur Stock Exchange, which is referred to as the *Bursa Malaysia*. In Malaysia, we believe that the agency theory intersects with institutional theory because the management's incentive for providing CSR information is partly politically motivated. For example, State Banks are heavily influenced by government initiatives, and PLCs are aware of them prioritizing CSR-favoring directives when evaluating loan applications. Given that most of these policy changes affecting CSR practices of Malaysian PLCs came into effect around 2009, our study concentrates on the post-regulatory period of 2009–2015 as the appropriate period to examine the economic implications of their effect on the cost of debt of Malaysian listed firms prior to Malaysia achieving the developed country status.

Dhaliwal et al. (2012) examined the relationship between CSR (nonfinancial) disclosure and analyst forecast accuracy in an international setting. Their findings show that the issue of standalone CSR reports is related to lower analyst forecast error in a country that is stakeholder oriented. Our current study examines whether creditors incorporate CSR disclosures into their cost of debt estimation. Goss and Roberts (2011) showed that banks provide soft financing for companies with better CSR performance. CSR investments in Malaysia are mainly in the form of soft debt financing provided by the government and state-owned banks' involvement in promoting CSR as a national agenda. Hence, the private sector takes advantage of these financing schemes offered by the government to spur the nation's economic growth and achieve Vision 2020. The present study builds upon the work by Ye and Zhang (2011). They examined the relationship between CSR investments relating to corporate philanthropy and debt financing costs in China, another emerging market. Unlike Ye and Zhang (2011), who used the risk mitigation theory, this study draws on both legitimacy and media agenda setting theories to explore how the cost of debt is reduced due to corporate governance and the winning of awards, which can lead to a reduction in business and financing risk.

Few studies in Malaysia have examined the role of the state in promoting CSR practices (Amran and Susela Devi 2008). Their study, using institutional theory, provides empirical evidence of the government's role in promoting the commitment to CSR initiatives under Agenda 21. Furthermore, the study by Subramaniam et al. (2016) examined the impact of government-led CSR policies and regulations on the liquidity of shares. They examined the relationship between different types of blockholdings and the liquidity of shares in Malaysia. The corporate governance variables of interest in their study were ownership structure, board size and institutional ownership. Their findings show that non-governmental institutional blockholdings improve the liquidity of shares, which also

interact with CSR disclosures to affect share liquidity. These authors did not investigate the impact of CSR disclosures on the cost of debt.

Sengupta's (1998) study was based on the notion that lenders and underwriters consider a firm's disclosure policy to estimate bond default risk. The sample in Sengupta's (1998) study was drawn from Standard & Poor's rating of accounting quality, suggesting that firms that consistently make timely and informative disclosures are perceived to have lower default risk. Our study differs from Sengupta (1998) in that we proxy winning CSR awards as value relevant in reducing information risk between creditors and PLCs.

Kansal et al. (2014) studied the determinants of CSR reporting in India, and one of the variables in their study was corporate social reputation. Their study highlights that corporate reputation, measured as awards and certifications, influences a company's legitimacy. Hence, they concluded that winning CSR awards can provide a significant reputation boost for a firm (Kansal et al. 2014). Furthermore, they suggested that highly reputable companies have positive stock returns and an adverse social risk. Their results also indicate that non-award-winning firms disclose significantly fewer CSR disclosures than award-winning firms. However, the study by Kansal et al. (2014) did not find any results to link CSR disclosures and risk, which is proxied by financial leverage. Considering the initiation of the MPs, the present study investigates the relationship between CSR disclosures and the cost of debt financing, a proxy for the reduction in business risk. Our study extends the study by Kansal et al. (2014) in two ways: First, we explore the moderating effect of corporate reputation (a non-financial attribute) affecting CSR disclosures on the cost of debt. Second, we investigate the link between corporate governance variables, the board size and types of institutional ownership as determinants of CSR disclosures. Bliss and Gul (2012) considered the relationship between politically connected Malaysian firms and debt cost. These authors highlighted that only a few studies had investigated the cost of debt and the determinants of debt pricing. Hence, we examined the determinants of CSR disclosure in an emerging economy setting and its relationship with the cost of debt capital.

The remainder of this paper is divided into four sections. Section 1 summarizes the Malaysian institutional setting, Section 2 reviews the literature and develops hypotheses, Section 3 discusses the research methodology, Section 4 presents the results and Section 5 is the conclusion section.

*The Malaysian Institutional Setting and the Development of Corporate Social Responsibility*

In 2007, Bursa Malaysia introduced the CSR Framework as part of its listing requirements for Malaysian PLCs (see note 1). This framework consists of guidelines that support triple bottom line reporting, which encourages Malaysian PLCs to engage in CSR activities and to disclose the details to all its stakeholders. The CSR Framework under the focal area of "*marketplace*" also incorporates good corporate governance mechanisms such as board independence and independent directors at board-level sub-committees, such as the audit committee. Paragraph 10 of MASB 1, *Presentation of Financial Statements* (now part of Financial Reporting Standard (FRS) 101), has provisions permitting "environmental reports and value-added statements" if they can help users in decisions about the allocation of their scarce economic resources. Additionally, FRS 137[2] (formerly MASB 20) requires disclosures when contingent liabilities and assets are recognized on a firm's Balance Sheet that are related to environmental and employment regulation violations. Consequently, given this regulatory pressure, the boards of Malaysian PLCs have a heightened responsibility to consider the importance of legitimacy in managing the magnitude of their companies' societal and economic risk by providing CSR disclosures.

## 2. Theoretical Framework and Hypothesis Development

### 2.1. Media Influence, CSR Disclosures and the Cost of Debt

This study utilizes the media agenda setting and legitimacy theories to explain the role of media in corporate social reporting. The institutional theory put forth by DiMaggio and Powell (1983) suggests that firms, in their quest for legitimacy, are subject to isomorphic

pressure that, over time, produces increasing similarity among peer organization disclosures. Lee and Hutchison (2005) stated that organizations operate within the boundaries of legitimacy set by society, meaning that adhering to both formal and informal requirements of a society is seen as a means for achieving legitimacy. Formal aspects include laws and regulations, whereas informal societal influences include publicity and public pressure on organizational legitimacy. Public pressure can arise from the press highlighting winners of CSR awards in media coverage. The theory states that a relationship exists between the relative importance given by the media to a particular topic and the degree of salience this topic has to the public (Ader 1995). The authors of institutional theory took the view that the media plays an essential role in developing social norms based on shaming non-compliant firms (Skeel 2001), which we argue brings isomorphic pressure on all Malaysian PLCs to mimic the behavior of leading CSR-disclosing firms. DiMaggio and Powell's (1983) institutional theory can be used to explain Malaysia's media agenda setting process that attempts to bring about greater homogeneity in voluntary CSR reporting. Hence, we argue that the legitimizing process of winning CSR awards results in a perception in society that the company is a standard setter in sustainable practices that spurs other companies within the same sectors to initiate similar CSR activities, which in turn increase their CSR disclosure levels (Lindblom 1994; Hambrick et al. 2004).

Using institutional theory, we argue that disclosures of social and environmental performance that go beyond mandatory (regulated) disclosures are used by companies to differentiate themselves from other competitors. Over the long run, mimetic isomorphic pressure brings about greater homogeneity in CSR reporting in developing countries such as Malaysia. Barnett (2007) showed that proactive CSR initiatives undertaken by firms to meet societal needs aid in acquiring legitimacy, which increases firm value and sustainability in the long term. Thus, we believe that Malaysian PLCs that have not won CSR awards can be considered less legitimate towards conforming to the country's social norms that are strongly entrenched in Malaysia, which is a high-power-distance collective society (https://geert-hofstede.com/malaysia.html, accessed on 20 September 2022). In such a society, media influence shows the importance of fulfilling the social contract and being legitimate in the eyes of society, as pointed out by Lindblom (1994). Hence, CSR awards are designed to improve PLCs' loyalty to the community, including present and future customers, which we believe can help companies increase their access to product and capital markets locally and globally.

Aerts and Cormier (2009) referred to the media as an 'institutional intermediary' whereby it creates public awareness of issues relating to social and environmental importance to society (Deegan et al. 2002). Fombrun (2005) noted that CSR awards tend to draw media attention to the winners, creating intangible benefits such as corporate reputation and customer goodwill, which we believe can be highly valuable to family-owned companies in Malaysia. Du and Vieira (2012) showed that winning CSR awards from third parties gives a firm creditability for its CSR activities and practices to stakeholders and investors. Market efficiency assumes that it is costless for investors to receive and process the signalling effect of winning awards. Hence, companies need to mimic the CSR disclosures of those favored companies identified for CSR awards. Consequently, we argue that this publicity of winning CSR awards acts as a first-move advantage to firms within the same industry because it helps to lower the cost of debt when borrowing from state-controlled banks in Malaysia.

The media-based 'shaming' process results in salient effects that influence investors and creditors' behaviors that make the conditions more stringent to receive funds for non-CSR award winners[3], particularly from state-controlled banks, and we believe PLCs are aware of such CSR-favoring directives from the government to state banks[4]. Fombrun (1996, p. 57) described 'reputation' as a strategic asset that "produce[s] tangible benefits such as premium prices for products, lower cost of capital and labour, improved loyalty from employees, greater latitude in decision making, and a cushion of goodwill when crises hit." The institutional-theory-based social impact hypothesis states that satisfying

stakeholders helps improve a firm's reputation, and it influences its financial performance (Roberts and Dowling 2002). Neu et al. (1998) argued that management tends to favor releasing non-financial data about social and environmental issues compared to financial information disclosures, as the former can be "tailored to public impressions" to give legitimacy to a firm's activities. Hence, firms might mimic other well-known Malaysian PLCs and respond to societal pressure by reporting on social and environmental issues to increase their legitimacy (Deephouse and Carter 2005; Kansal et al. 2014). Prior studies have shown that reporting CSR initiatives increases transparency and enhances firms' corporate reputation (Zulkifli and Amran 2006). A Malaysian study by Sadou et al. (2017) shows that winning CSR awards is the main factor influencing CSR disclosures. Hence, our study uses the winning of CSR awards as an institutional proxy measure to capture the level of isomorphism or homogeneity in CSR transparency and accountability, and it examines its effect on the cost of debt. Hence, the hypotheses are stated as follows:

**Hypothesis 1 (H1a).** *Ceteris paribus, there is a negative relationship between CSR disclosures and the cost of debt.*

**Hypothesis 1 (H1b).** *Ceteris paribus, there is a negative relationship between winning awards and the cost of debt.*

*2.2. Board Size, CSR Disclosures and the Cost of Debt*

Studies by Jo and Harjoto (2012) and Cai et al. (2012) establish that better-governed corporations are pre-disposed to engaging in better CSR practices than poorly governed companies. Ho's (2005) holistic framework views CSR as an integral part of the company's governance regime, which focuses on non-financial risks and suggests that companies develop procedures to mitigate such risks. The objective of governance mechanisms and the voluntary disclosure of CSR activities are more likely to converge on the issue of increasing accountability and transparency to investors and other firm stakeholders. Cong and Freedman (2011) observed that firms with effective corporate governance show better accountability by disclosing more environmental disclosures. Hence, the practice of CSR disclosures in companies' annual reports or in standalone reports prepared in accordance with the Global Reporting Initiative and Sustainability Reporting Guidelines can be viewed as directors discharging their fiduciary responsibilities and accountability to all their stakeholders.

The board of directors are seen as vital to a company and as monitors and advisers to management. Lipton and Lorsch (1992) proposed that the maximum number of directors on a board should be limited to ten directors, as increasing the board size may result in communication and coordination problems that result in poor decision making and lower quality of financial reporting and transparency. The notion of having small, effective and focused boards is also supported by Yermack's (1996) US study. For example, Yermack (1996) showed that small boards have a higher market valuation, measured using Tobin's *Q*. In contrast, the study by Coles et al. (2008) shows that complex firms and those relying on debt financing have larger boards, which enable them to obtain advice due to their diverse knowledge and expertise. A Malaysian study by Abdifatah Ahmed (2013) examines the level of CSR disclosures over a period of time that underwent changes in the regulatory and business environment, and it shows that there is a positive relationship between board size and the level of CSR disclosures. A study by Sadou et al. (2017) examines the level and determinants of CSR disclosures between 2011 and 2014 and found that board size, ownership concentration, independent directors and return on assets influence the level and quality of CSR disclosures. Our study examines the association between board size and the cost of debt. We posit that the board monitors and advises management on sustainability practices that help to improve the level of CSR disclosures, which in turn affects the cost of debt. Therefore, we hypothesize that:

**Hypothesis 2 (H2).** *Ceteris paribus, board size is associated with the cost of debt.*

*2.3. Political Connections and the Cost of Debt*

In this study, politically connected firms are identified by the criteria used by Subramaniam and Sakthi (2022), Bliss and Gul (2012) and Johnson and Mitton (2003). The criteria used are when either the directors or major shareholders have informal ties with key government officials, such as with Khazanah, the Ministry of Finance, Bank Negara Malaysia or government-related agencies in which the government has an interest due to its financial or legal exposure. Lin et al. (2015) noted that companies form political connections with governments to reap benefits such as lower local taxation (Faccio 2010) and to obtain privileged access to debt financing from government-controlled banks (Fan et al. 2008; Li et al. 2008). Malaysia is also characterized by the presence of politically connected companies and state-controlled banks (Mohd Ghazali and Weetman 2006). Cheng et al. (2017) examined the relationship between political connections and CSR disclosures in heavily polluted Chinese industries. Their results indicate that politically connected companies promote more environmental information but are seen as increasing the quantity rather than the quality of environmental information. Hence, stakeholders find that it becomes difficult to make effective decisions about a firm's CSR due to the lower quality or transparency of environmental disclosure. Chaney et al. (2011) found that politically connected firms, when compared to non-connected firms, report lower quality earnings, indicating the presence of higher information asymmetry in politically connected companies. Following Chaney et al. (2011) and Cheng et al. (2017), we argue that information asymmetry leads to lower levels of information disclosure, which in turn affects the cost of debt. Therefore, we hypothesize that:

**Hypothesis 3 (H3).** *Ceteris paribus, there is a positive relationship between politically connected companies and the cost of debt.*

*2.4. Government Ownership and the Cost of Debt*

The major institutional investors in Malaysia are the Employees' Provident Fund (EPF), Lembaga Tabung Haji (formerly known as the Pilgrimage Management and Fund Board) and Permodalan Nasional Berhad (Malaysia's biggest fund management agency). Wahab et al. (2007) found that government institutional investors such as the EPF play an important role in promoting good corporate governance practices, improving minority shareholder interest and increasing the level of financial and non-financial disclosures. According to Marinetto (1998), public institutional investors impose a fiduciary duty based on a social–ethical dimension to ensure that they do not undertake activities that cause harm to the wider society. The Asian financial crisis of 1997–1998 gave prominence to the role of institutional investors such as the public employee pension fund, which is a government-related institution (referred to as the Employees' Provident Fund or the EPF) that invests employees' pension contributions. Government ownership plays a significant role in ensuring that boards fulfill their fiduciary duty of reducing agency costs by increasing voluntary disclosures (Hawley and Williams 1997). Therefore, we examine government institutional ownership as a factor that determines the level of CSR disclosures. Hence, via better governance and disclosure, the quality of a board may have a material impact on the cost of debt capital. Moreover, winning CSR awards provides greater credibility for non-financial disclosures, which in turn may help reduce information asymmetry between the company and its creditors.

Regarding the institutional theory perspective, governments act in their capacity as societal institutions. As a result, they are able to exercise coercive power over PLCs via regulations and their enforcement (DiMaggio and Powell 1983; Scott 2001). The evidence from the study by Ntim and Soobaroyen (2013) suggests that companies with high government ownership keenly lobby for government support by engaging in increased CSR disclosures. This support enables a company to seek legitimacy of its business operations

and to assist in accessing additional resources such as subsidies/tax holidays, which can enhance a PLC's financial performance. Furthermore, Eng and Mak (2003) stated that the ownership structure of a company influences the monitoring level, thereby affecting voluntary disclosures. In contrast, Cressy et al. (2012) observed that countries with poor governance practices and weak enforcement have higher levels of corruption. Thus, high government ownership can lead to poor or less transparent CSR practices as well. However, evidence from Eng and Mak (2003) and Khan et al. (2013) indicates that, in Singapore and Malaysia, government ownership is positively related to CSR practices. Shailer and Wang (2015) studied the impact of the government controlling ownership on the cost of debt of Chinese listed companies. Their results indicate that firms under government control have a lower cost of debt compared to corporations under private control because government ownership is beneficial when companies show signs of financial distress, likely to borrow funds from state banks. Hence, our fourth hypothesis is stated as follows:

**Hypothesis 4 (H4).** *Ceteris paribus, as the percentage of government shareholdings becomes higher, the cost of debt becomes lower.*

*2.5. Non-Governmental Institutional Ownership*

Other non-governmental institutional investors are defined to include holdings by banks, investment companies, mutual funds, security companies and insurance companies. A study by Subramaniam et al. (2016) posits that other non-governmental investors play an effective role in monitoring and participating in board decisions, as they intend to maximize firm value in the long term by reducing information asymmetry that affects the bid–ask spreads of equity shares of Malaysian PLCs. In this study, we argue that the firm value can be maximized by a lower cost of debt financing, which is likely more important to Malaysian PLCs given the prevalence of closely held companies that prefer debt over equity capital (Lefort and Walker 2007). Thus, Malaysian PLCs that desire investments from these non-governmental institutional investors attempt to seek a reputation for being good stewards and provide more voluntarily CSR disclosures, which leads to a lower cost of debt financing. Hence, our fifth hypothesis is stated as follows:

**Hypothesis 5 (H5).** *Ceteris paribus, as the percentage of non-government shareholdings becomes higher, the cost of debt becomes lower.*

*2.6. Audit Committee Independence and the Cost of Debt*

In the governance process, the audit committee consisting of independent directors is associated with the effective monitoring of the financial accounting process and the reliability of financial reports. Prior research indicates that this enhances disclosure quality in financial reports (Ho and Wong 2001; Forker 1992). Furthermore, the presence of an audit committee reduces information asymmetries between management and stakeholders and, at the same time, leads to a reduction in agency costs. Li et al. (2012) studied the relationship between audit committee characteristics and the extent of intellectual capital disclosures in the UK. Their findings show that audit committee independence does not influence the level of intellectual capital disclosures.

Anderson et al. (2003) found that a fully independent audit committee is associated with a significantly lower cost of debt financing, because the creditors demand that the audit committees be independent with an increase in the company's debt-to-assets ratio or leverage. The rationale is that creditors demand more monitoring to rely on the reliability of the company's financial reports (Klein 2002). However, studies by Anderson et al. (2003) and Lorca et al. (2011) on Spanish companies did not find any association between audit committee independence and the cost of debt. Klein's (2002) findings also show that there is no association between audit committee independence and the demand for accounting information from creditors. Hence, to test this association, our sixth hypothesis is stated as follows:

**Hypothesis 6 (H6).** *Ceteris paribus, as audit committee independence becomes higher, the cost of debt becomes lower.*

*2.7. Moderating the Role of CSR Disclosures and CSR Awards on the Cost of Debt*

Antunovich et al. (2000) showed that companies included in the ranks of "*America's Most Admired Companies*" in Fortune magazine had significantly higher abnormal returns because of their elevated reputation. Bebbington et al. (2008) showed that companies may use sustainability disclosures to reduce their reputation risk. Hence, in the Malaysian capital market, by studying the effect of CSR awards on the legitimizing process, we believe that awards might also circuitously influence the cost of debt because of the dominance of state-controlled banks[5] and government-controlled institutional investors. Stiglitz (1993) noted that the objective of state-owned banks is to channel resources to socially profitable initiatives. The economic theory of institutions suggests that state-owned banks are created to address market failures whenever the social benefits of state-owned enterprises exceed their costs (Atkinson and Stiglitz 2015). Thus, with the dominance of state-controlled banks, we argue that the significance of increasing CSR disclosures and winning CSR awards by Malaysian PLCs is associated with a lower cost of debt. Deegan and Carroll (1993) noted that winning CSR awards indicates to the market that a firm is socially responsible and that it is able to continue to sustain its CSR performance.

Sengupta (1998) found that firms with high disclosure ratings from financial analysts benefit from lower interest rates. For this study, using the media agenda setting theory as a premise, we propose the hypothesis that the announcement of CSR award winners has a signalling effect in capital markets, which in turn may help reduce the cost of the debt capital of Malaysian PLCs. We choose to examine the cost of debt because we expect that companies receiving corporate responsibility awards first improve their creditability and reputation in the eyes of the public through media coverage that comes from winning the CSR awards. Second, PLCs may reduce the cost of borrowing from state-controlled banks that implicitly favor firms seen as more legitimate because of winning government-sponsored CSR awards.

In Malaysia, the Star newspaper takes a prominent role in promoting and monitoring CSR activities in the private sector and announces CSR award winners. Stakeholders perceive winning awards as a form of legitimacy (Ettenson and Knowles 2008). Consequently, reputation acts to reduce the risk of doing business with the firm, and it is easier to raise debt financing from state-controlled banks. Being a member of the Billion Ringgit Club (*BRC*)[6] and the influence of the media in announcing winners acts as a channel to obtain cheaper financing.

Firms with higher (lower) levels of CSR disclosures and stronger governance face lower (higher) costs of borrowing. These concerns include increased risk or the ability of the firm to repay its debt. In order to ensure that the Malaysian government's goals on national development and sustainability are met, institutional investments are vital to the development of the capital market (Singh 1991). Therefore, having increased CSR disclosures and winning CSR awards can influence government and non–governmental institutional investors to invest more in socially responsible firms. Thus, we propose that winning CSR awards moderates the level of CSR disclosures, and we state the following hypothesis:

**Hypothesis 7 (H7).** *CSR awards moderate the relationship between CSR disclosures and the cost of debt.*

**3. Research Methodology**

*3.1. Construction of the CSR Index*

CSR research in Malaysia has been based on content analysis or self-constructed disclosure indices (Said et al. 2009; Haniffa and Cooke 2002), but they have not been based on the best practices (Subramaniam et al. 2016). The media, particularly *The Star* newspaper

and its CSR award, plays an influential role in promoting the sustainability agenda among Malaysian PLCs. The awards are assessed based on the Bursa Malaysia CSR framework, which emphasizes the marketplace, workplace, environment and community involvement of Malaysia PLCs. *The Star* newspaper, ICR Malaysia and the Security Commission of Malaysia designed the survey instrument. A similar survey instrument was used to construct the index that incorporated the Millennium Development Goals into the 10th Malaysian Plan (2011–2015) as part of its national development strategy. Therefore, this is a better scoring instrument to measure the quality of CSR disclosures in Malaysian PLCs. Thus, we use the criteria used by StarBiz-ICR Malaysia to develop our CSR Index.

We follow the methodology used by Subramaniam et al. (2016) to develop our CSR index. The CSR Index is calculated based on scores assigned ranging from 0 to 3 for the following sections: (1) the structuring of community activities by the company; (2) community consultation and investment; (3) workplace: employment conditions; (4) workplace health and safety standards; (5) workplace: consultation practices; (6) environmental resources, emissions and waste standards; (7) environmental support for biodiversity; and (8) standards for reporting on these activities (see Appendix A). This is further sub-classified as social and environmental disclosures. Additionally, we calculate the number of words describing the firms' CSR and environmental policies in each company's annual report. The number of words that describe their CSR and environmental policies in companies' annual reports is counted. For each company, the median CSR score for that particular industry is subtracted from the CSR raw scores of each company that belongs to that particular sector, which represents either below or above the median of each sector and is labelled Industry_CSR. The CSR_INDEX is constructed by adding the Industry_CSR to the CSR_Score, which is the number of word counts describing it.

Quantitative information is given more weight than mere narrative disclosures, as this goes to establish management's creditability to stand by the statements that they make in annual reports. We justify using these CSR variables for two reasons. First, Malaysian companies are still in the development stage of the CSR-reporting framework (Buniamin 2010). Second, Bursa Malaysia's CSR 2007 Framework promotes CSR disclosures by companies in their annual reports, and the number of words might capture the effort by the company to differentiate itself (i.e., to show its willingness to signal greater CSR disclosures).

*3.2. Sample Selection*

The information is obtained from annual reports on the Bursa Malaysia website. The corporate governance and financial accounting variable data are collected for fiscal years 2009 to 2015. Data are extracted from the OSIRIS financial database where they cannot be obtained from the Bursa Malaysia website. The sample consists of 104 firms or 728 firm year observations, comprising members of "*The Edge Billion Ringgit Club and Corporate Awards*". Financial institutions that are subject to different reporting and regulatory requirements under Bursa Malaysia's rules are excluded from the sample. To be included in *BRC*, companies must have at least RM1 billion in sales or market capitalizations. The *BRC* companies are portrayed as exemplars of stewardship for the broader society. The main reasons for examining *BRC* companies are, first, they are Malaysia's biggest and best performing companies listed on Bursa Malaysia. As such, the underpinning assumption is that these companies, in order to curry favor from state-controlled banks, engage in greater voluntary CSR disclosures. Second, these awards act as the benchmark for Malaysia PLCs to promote legitimacy among the wider society. Third, prior research has also shown that the level of disclosure is positively related to the size of a company (Holland and Foo 2003), and those belonging to *BRC* are the largest Malaysian PLCs. Hence, they should be the ones to provide greater CSR disclosures, and they want to curry favor from banks.

*3.3. Measurement of the Dependent, Independent and Control Variables*

3.3.1. Dependent Variable

The dependent variable, the cost of debt (COD_INT)), is measured as the firm's interest expense divided by its average short-term and long-term debt during the year. The interest expense for the year is disclosed in the income statement, and the short-term and long-term debt are disclosed in the balance sheet of the financial statements in the annual reports. Bliss and Gul (2012) and Pittman and Fortin (2004) used a similar measure of the cost of debt.

3.3.2. Independent Variables

CSR_INDEX

The CSR index (CSR_INDEX) is a self-constructed index that is obtained by calculating the total scores given to each firm under the StarBiz awards criteria. Thereafter, the median adjusted score for each firm within the industry is added to the number of words describing the CSR policy of the firm.

Board size (BO_SIZE) is measured as the number of directors on the board (Abdifatah Ahmed 2013). Audit committee independence (AC_IND) is measured as a percentage of independent directors on the audit committee (Said et al. 2009).

We include two explanatory variables of ownership concentration, namely government institutional ownership (GOV_INST) and non-governmental institutional ownership (OTH_INST). We follow Subramaniam et al. (2016) in measuring government institutional ownership (GOV_INST) as the percentage of company shares held by government-related institutional investors, directly or indirectly, such as EPF, LTAT, PNB, LTH and PERKESO/SOCSO[7]. Other non-government institutional ownership (OTH_INST) is measured as the percentage of shares held by all other institutional investors (excluding EPF, LTAT/, PNB, LTH and PERKESO/SOCSO) holding at least 5 percent of outstanding shares.

We follow the studies by Subramaniam and Sakthi (2022), Bliss and Gul (2012) and Johnson and Mitton (2003) to measure politically connected firms (POL_CON) as either an indicator variable that equals 1 if the company is politically connected and 0 otherwise. The CSR award (CSR_AWARDS) is also a dummy variable that is a proxy for whether a company has applied and received a CSR award from StarBiz or any other reputable organizations, such as the Association of Certified Chartered Accountants and the Prime Minister's CSR Awards organized by the Ministry of Women, Family and Community Development. Particularly, a company's reputation is operationalized as a winner of CSR awards.

Control Variables

The firm size (Size) is measured as the natural log of total assets. The study uses total assets as a proxy for a firm's size, which has been widely used by other CSR researchers (e.g., Eng and Mak 2003; Haniffa and Cooke 2005; Mohd Ghazali and Weetman 2006). Creditors' power (CRED_POWER) is a proxy for financial leverage and is measured as the total liabilities as a percentage of the total assets. Highly leveraged firms tend to have heightened monitoring by creditors (Jensen and Meckling 1976). Consequently, leveraged firms may have a higher level of voluntary disclosures to meet the demands of not only the creditors but also all stakeholders. Growth (GROWTH) is measured as Tobin's Q. Campbell (2007) suggested that companies with weak financial performance, proxied by low profitability and low growth rates, tend to engage less in CSR activities. Appendix B describes all the variables, definitions, labels and measurements.

*3.4. Empirical Model Specifications*

We use White's test to handle heteroskedasticity correlations in the residuals. All observations lying below the 1st and above the 99th percentiles of the distribution are censored for each test variable. The regression analysis tests the relationship between the explanatory variables and the cost of debt. The result of the fixed effects model is compared to the random effects model using Hausman's test to test for the effects of homogeneity,

and based on that, the fixed effects model is selected. Regression Model 1 and the expanded regression Model 2 with interaction are used for testing, which are as follows:

**Model 1**:

$$\text{COD\_INT}_{it} = \alpha_0 + \alpha_1\,\text{CSR\_INDEX} + \alpha_2\text{CSR\_AWARDS} + \alpha_3\text{POL\_CON} + \alpha_4\text{GOV\_INST} + \alpha_5\text{OTH\_INST} +$$
$$\alpha_6\text{BO\_SIZE} + \alpha_7\text{AC\_IND} + \alpha_8\text{ROA} + \alpha_9\text{GROWTH} + \alpha_{10}\text{LnSIZE} + \tag{1}$$
$$\alpha_{11}\text{AGE\_FIRM} + \alpha_{12}\text{CRED\_POWER} + \varepsilon$$

**Model 2**:

$$\text{COD\_INT}_{it} = \alpha_0 + \alpha_1\,\text{CSR\_INDEX} + \alpha_2\text{CSR\_AWARDS} + \alpha_3\text{CSR\_INDEX*CSR\_AWARDS} +$$
$$\alpha_4\text{POL\_CON} + \alpha_5\text{GOV\_INST} + \alpha_6\text{OTH\_INST} + \alpha_7\text{BO\_SIZE} + \alpha_8\text{AC\_IND} + \alpha_9\text{ROA} + \tag{2}$$
$$\alpha_{10}\text{GROWTH} + \alpha_{11}\text{LnSIZE} + \alpha_{12}\text{AGE\_FIRM} + \alpha_{13}\text{CRED\_POWER} + \varepsilon$$

## 4. Results and Discussion

### 4.1. Preliminary Analysis

Table 1, Panel A shows the criteria used in the sample selection for 2009 to 2015, and Table 1, Panel B shows the sectors analyzed. A majority of the companies are from the industrial sector (31.7%), followed by industrial production (23.1%), consumer products (13.5%), plantation (9.6%), construction/infrastructure (7.7%), the property/hotel sector (5.8%), technology (4.8%), retail (1.9%), oil and energy (0.97%) and the airline/shipping (0.97 %) sectors.

**Table 1.** Sample selection and distribution by industrial sectors.

| Panel A: Data and Sample | |
| --- | --- |
| **Selection Criteria** | **No. of Companies** |
| Billion Ringgit Club (*BRC*) companies as of 31 March 2010 | 163 |
| Less: | |
| Banking and Financial Services | 25 |
| Companies ceasing to be part of the *BRC* due to delisting or merging | 34 |
| Usable sample | 104 |

| Panel B: Industry Distribution | | | | |
| --- | --- | --- | --- | --- |
| **Sector** | **Industry Category** | **No. of Companies** | **Frequency** | |
| | | | **Absolute** | **Relative (%)** |
| 1 | Trading and services | 33 | 231 | 31.7 |
| 2 | Properties | 6 | 42 | 5.8 |
| 3 | Industrial Production | 24 | 168 | 23.1 |
| 4 | Plantation | 10 | 70 | 9.6 |
| 5 | Consumer Products | 14 | 98 | 13.5 |
| 6 | Technology | 5 | 35 | 4.8 |
| 7 | Construction/Infrastructure | 8 | 56 | 7.7 |
| 8 | Oil and Energy | 1 | 7 | 0.97 |
| 9 | Airline/Shipping | 1 | 7 | 0.97 |
| 10 | Retail | 2 | 14 | 1.9 |
| | Total | 104 | 728 | 100 |

Note: The sample distribution of 728 company year observations covers the period of 2009 to 2015.

Table 2 shows the descriptive statistics of the variables. It shows that the CSR_Index has a mean (median) of 64.37% (55%). The average government ownership (GOV_INST) level in our sample is 12.11%. Regarding corporate governance, the audit committee consists of 90% independent directors (AC_IND). Furthermore, only 21% of the companies that belong to *BRC* have won an award for their CSR practices, and 32.7% of the sample firms have political connections. The cost of debt shows that, on average, the companies' interest rate is only 5.2%.[8]

**Table 2.** Descriptive statistics of dependent and independent variables.

| Variables | Mean | Median | Std. Dev. | Minimum | Maximum |
|---|---|---|---|---|---|
| COD_INT | 0.067 | 0.052 | 0.057 | 0.023 | 0.087 |
| CSR_INDEX | 64.371 | 55.000 | 53.191 | 32.00 | 85.750 |
| AC_IND | 0.900 | 1.000 | 0.140 | 1.000 | 1.000 |
| BO_SIZE | 8.279 | 8.000 | 1.804 | 7.000 | 9.000 |
| GOV_INST | 0.121 | 0.069 | 0.178 | 0.000 | 1.000 |
| OTH_INST | 0.114 | 0.089 | 0.112 | 0.000 | 1.000 |
| POL_CON | 0.327 | 0.000 | 0.469 | 0.000 | 1.000 |
| CSR_AWARDS | 0.209 | 0.000 | 0.411 | 0.000 | 1.000 |
| AGE_FIRM | 35.491 | 33.000 | 22.037 | 20.000 | 45.000 |
| GROWTH | 1.234 | 0.730 | 1.791 | 0.000 | 14.000 |
| ROA | 0.149 | 0.076 | 0.558 | 1.180 | 8.300 |
| LNSIZE | 14.96 | 14.70 | 1.22 | 12.550 | 18.58 |
| CRED_POWER | 1.339 | 0.854 | 2.341 | 0.040 | 27.120 |
| N | 728 | 728 | 728 | | |

Table 3 presents the Pearson correlation coefficients, which show that COD_INT is significantly negatively correlated with BO_SIZE, OTH_INST and AGE_FIRM. Additionally, CSR_Index is significantly positively correlated with CSR_AWARDS, BO_SIZE and GOV_INST. Political connection (POL_CON) is negatively and significantly related to CSR_INDEX and audit committee independence (AC_IND) but is positively and significantly related to board size (BO_SIZE). Hence, the univariate statistic results provide support for our hypotheses that the level of CSR disclosures is influenced by ownership type and corporate governance mechanisms.

**Table 3.** Correlation analysis (Pearson correlations coefficients).

| | COD_INT | CSR_INDEX | CSR_AWARD | BO_SIZE | AC_IND | POL_CON | GOV_INST | OTH_INST | ROA | GROWTH | LNSIZE | AGE_FIRM | CRED_POWER |
|---|---|---|---|---|---|---|---|---|---|---|---|---|---|
| COD_INT | 1 | | | | | | | | | | | | |
| CSR_INDEX | −0.133 | 1 | | | | | | | | | | | |
| CSR_AWARD | 0.007 ** | 0.071 * | 1 | | | | | | | | | | |
| BO_SIZE | −0.039 ** | 0.022 ** | 0.203 | 1 | | | | | | | | | |
| AC_IND | 0.045 ** | −0.007 * | −0.074 * | −0.027 ** | 1 | | | | | | | | |
| POL_CON | 0.108 | −0.071 * | 0.211 | 0.080 * | −0.074 * | 1 | | | | | | | |
| GOV_INST | −0.003 ** | 0.054 ** | 0.253 | 0.171 | −0.182 | 0.395 | 1 | | | | | | |
| OTH_INST | −0.077 * | −0.039 ** | 0.157 | 0.269 | −0.128 | 0.059 * | 0.066 * | 1 | | | | | |
| ROA | 0.099 * | −0.193 | −0.013 ** | −0.007 * | −0.037 ** | −0.003 ** | −0.055 * | 0.133 | 1 | | | | |
| GROWTH | −0.202 | −0.101 | 0.239 | 0.005 ** | −0.179 | 0.023 ** | −0.034 ** | 0.088 * | 0.120 | 1 | | | |
| LNSIZE | 0.204 | 0.031 ** | 0.270 | 0.240 | −0.016 ** | 0.250 | 0.317 | 0.246 | 0.008 * | −0.157 | 1 | | |
| AGE_FIRM | −0.112 | 0.160 | −0.065 * | −0.159 | −0.030 ** | −0.035 ** | −0.076 * | −0.104 * | −0.096 * | 0.040 ** | −0.082 | 1 | |
| CRED_POWER | 0.117 | 0.007 * | 0.064 * | 0.029 ** | −0.058 * | −0.010 ** | 0.011 ** | −0.003 ** | −0.007 * | 0.002 ** | 0.070 | −0.054 ** | 1 |

All variables are defined in Appendix B. ** Significant at the 5% level, * Significant at the 10% level.

## 4.2. Estimation Results

Table 4 shows the results for Model 1, which is a regression of the explanatory variables on the cost of debt (COD_INT). Hypothesis H1a states that there is a negative relationship between CSR disclosures and the cost of debt. Model 1 shows that the CSR disclosures (CSR_INDEX) variable is negatively and significantly ($\alpha = -0.001$, $p < 0.05$) correlated with COD_INT. The results support prior evidence that, when there are high-quality financial and social disclosures, it helps to reduce the cost of debt capital by decreasing information asymmetry between PLCs and creditors (Cuadrado-Ballesteros et al. 2016), which implies that, as the level of CSR disclosures becomes higher, the cost of debt becomes lower. Hence, we conclude that Hypothesis H1a is supported. Hypothesis H1b states that there is a negative relationship between winning CSR awards and the cost of debt. However, Model 2's results show a positive relationship between CSR_AWARDS and the cost of debt. Thus, our results suggest that winning CSR awards does not increase the credibility of Malaysian PLCs' CSR disclosures, which provides evidence contrary to hypothesis H1b. Subramaniam et al. (2016) noted that creditors do not rely on CSR information provided by companies in emerging capital markets, as the information has not been authenticated by a third party for accuracy.

**Table 4.** Regression results of Model 1 using COD_INT as a dependent variable, and Model 2 uses the moderator effect of CSR_AWARDS and CSR_INDEX.

| Variable | Model 1 | | Model 2 | |
|---|---|---|---|---|
| | Coefficient | t-Statistic | Coefficient | t-Statistic |
| Constant | −0.004 | −0.489 | −0.006 | −0.681 |
| CSR_INDEX | −0.001 | −3.360 *** | 0.000 | −1.566 |
| CSR_AWARDS | 0.001 | 0.910 | 0.006 | 2.772 ** |
| CSR_INDEX*CSR_AWARDS | | | 0.000 | −2.900 *** |
| BO_SIZE | −0.001 | −1.660 * | 0.000 | −1.440 |
| AC_IND | −0.001 | 0.952 | −0.001 | −0.303 |
| POL_CON | 0.003 | 2.503 ** | 0.003 | 2.689 *** |
| GOV_INST | −0.008 | −2.350 ** | −0.008 | −2.474 ** |
| OTH_INST | −0.017 | −3.407 *** | −0.016 | −3.256 *** |
| ROA | 0.003 | 2.962 ** | 0.004 | 3.225 *** |
| GROWTH | −0.002 | −5.205 *** | −0.002 | −5.589 *** |
| LNSIZE | 0.002 | 4.886 *** | 0.002 | 4.995 *** |
| AGE_FIRM | −0.001 | −2.322 ** | 0.000 | −2.138 ** |
| CRED_POWER | 0.001 | 2.885 *** | 0.000 | 2.921 *** |
| Period | Yes | | Yes | |
| Industry | Yes | | Yes | |
| $R^2$ | 0.156 | | 0.167 | |
| Adjusted $R^2$ | 0.135 | | 0.144 | |
| No. of observations | 728 | | 728 | |
| F-Ratio | 7.326 | | 7.456 | |
| Probability | 0.000 | | 0.000 | |

All variables are defined in Appendix B. *** Significant at the 1% level, ** Significant at the 5% level, * Significant at the 10% level.

Hypothesis 2 states that board size is associated with the cost of debt. Model 1's results indicate that BO_SIZE is negatively and marginally significantly related to the cost of debt proxy COD_INT ($\alpha = -0.000$, $p < 0.10$). This indicates that, as the board size decreases, the cost of debt decreases as well, which provides marginal support for Hypothesis H2. Prior studies have found similar evidence of the effectiveness of small boards. For example, Yermack (1996) and Anderson et al. (2003) found that a smaller board size positively influences corporate transparency. Upadhyay and Sriram (2011) found a negative association between board size and the cost of capital, measured as the weighted cost of capital. Hence, based on the disclosure literature, we conclude that smaller boards are more effective in monitoring CSR disclosures in Malaysian PLCs.

Hypothesis 3 states that there is a positive relationship between politically connected (POL_CON) companies and the cost of debt. The results of both models show a significant and positive relationship between PON_CON and COD_INT ($\alpha = 0.003$, $p < 0.05$ in Model 1 and $p < 0.10$ level in Model 2), indicating that POL_CON companies have a higher cost of debt. Bliss et al. (2011) found that the association between audit committee independence and the demand for higher quality audits are weaker for politically connected companies, and such firms are perceived to be of higher risk by auditors and lenders (Bliss and Gul 2012). Our findings show that audit committee independence (AC_IND) is positive but insignificant, indicating a lack of significance of audit committees in helping to lower transparency and information asymmetry. Hence, our results suggest that, when audit quality is weak, the credibility of CSR disclosures is low in politically connected companies, which results in a higher cost of debt in Malaysian PLCs. We further contribute to the study by Bliss et al. (2011) by examining the independence of the audit committee on CSR disclosures made by politically connected firms on the cost of debt, unlike the Bliss et al. (2011) study, which examines the independence of the audit committee on audit fees. Therefore, we conclude that there is support for hypothesis H3.

Hypothesis 4 states that, as governmental shareholdings become higher, the cost of debt becomes lower. The coefficient of GOV_INST is negative and significant ($\alpha = -0.008$,

$p < 0.05$). Shailer and Wang (2015) studied the impact of government-controlled ownership on the cost of debt of Chinese listed companies. Their results indicate that firms under government control have a lower cost of debt compared to corporations under private control, because government ownership is beneficial when companies show signs of financial distress. Subramaniam et al. (2016) found that government ownership and CSR disclosures act as substitute governance mechanisms that influence the liquidity of Malaysian PLCs. As a result, due to the substitution effect, we suggest that the cost of debt is reduced with government ownership. Hence, based on our findings and prior results, we conclude that there is support for Hypothesis 4.

Hypothesis 5 states that, as the percentage of non-governmental shareholdings becomes higher, the cost of debt becomes lower. The coefficient OTH_INST shows a negative relationship in both models ($\alpha = -0.017$ or $-0.016$, $p < 0.05$), implying that the shareholdings by non-governmental institutions reduce the cost of debt of Malaysian PLCs. The presence of OTH_INST reduces the agency cost with greater monitoring, which is also consistent with the results of Subramaniam et al. (2016). Hence, we conclude that hypothesis H5 is supported.

Hypothesis 6 states that, as the audit committee becomes more independent, the cost of debt becomes lower. The results show that AC_IND is negative but not significant. Therefore, hypothesis H6 is not supported. Model 2 in Table 4 includes the interaction terms CSR_INDEX*CSR_AWARDS to examine the moderating effect of winning CSR_AWARDS on the CSR_INDEX in indirectly affecting the cost of debt. Our interest is to establish whether the effect of winning CSR_AWARDS influences the perceptions of stakeholders through media visibility and news coverage that promote the authenticity of CSR reporting[9]. Hence, stakeholders expect higher CSR disclosures from companies who have won CSR awards to be more credible. Thus, hypothesis H7 states that CSR_AWARDS moderates the effect of CSR disclosures on the cost of debt due to the greater credibility of such disclosures, as vouched by the media.

Table 4, Model 2 shows that the coefficient CSR_INDEX*CSR_AWARDS is negative and significant ($\alpha = 0.001$, $p < 0.05$). However, now CSR_AWARDS is positive and significant ($\alpha = 0.006$, $p < 0.10$), which implies that, when a company wins an award, the cost of debt increases, because the market may believe that the PLCs overinvested in CSR activities to gain legitimacy (Barnea and Rubin 2010). The results of our study seem to be consistent with Boachie and Tetteh (2021) in Ghana, who found that lenders tend not to rely on CSR disclosures as a mitigating risk factor. The main concern by lenders is the possibility of overinvestment by management in CSR investments by diverting scarce resources (Barnea and Rubin 2010). This is further evidenced by the ROA being more positive and significant in Model 2 ($\alpha = 3.225$, $p < 0.001$) compared to Model 1 ($\alpha = 2.962$, $p < 0.001$), creating a conflict between profitability and legitimacy, as creditors perceive the firms to be at high risk due to overinvestment, causing an increase in the cost of debt. In Model 2, our findings indicate that CSR_INDEX is insignificant, but CSR_AWARDS is positive and significant ($\alpha = 2.772$, $p < 0.05$). Hence, winning an award is the reason for the moderating negative effect of CSR_AWARDS and CSR_INDEX on the cost of debt being significant ($\alpha = -2.900$, $p < 0.001$). Figure 1 visualizes the moderation effect shown in Table 4, Model 2.

According to Abdifatah Ahmed (2013), winning CSR awards is a strategy that Malaysian companies use to gain legitimacy as companies with the best practices. Examples of CSR awards include the Prime Minister's CSR award and ACCA MESRA (Malaysian Environmental and Social Reporting Award). Media coverage of firms winning CSR awards creates a perception amongst stakeholders of the firm's social responsibility (legitimacy) towards society and improves their reputation and branding of Malaysian PLCs. Furthermore, winning the awards has a signalling effect in terms of reducing information asymmetry by providing authenticity for CSR disclosures, which lowers the cost of debt. Thus, we believe that firms are motivated to win CSR awards by investing in CSR initiatives and activities to obtain legitimacy within society, expecting a lower cost of financing.

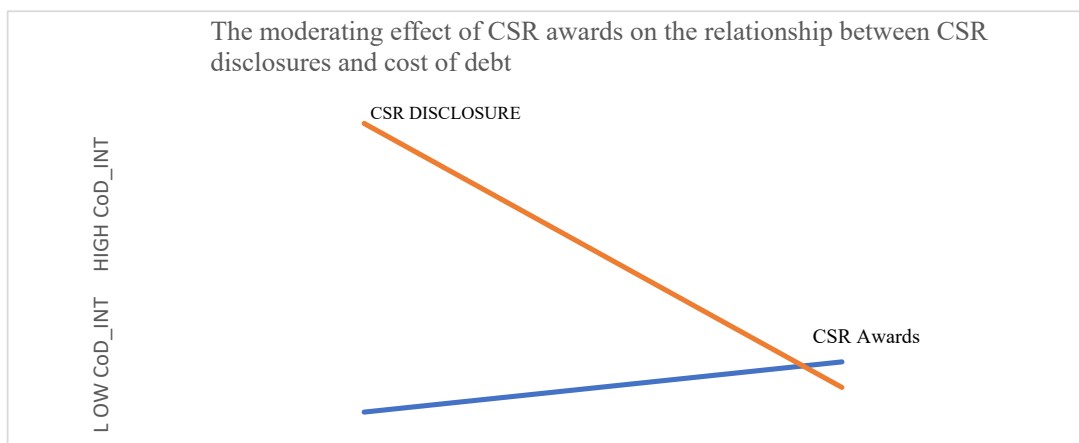

**Figure 1.** The moderating effect of CSR awards on the relationship between CSR disclosures and the cost of debt.

In Models 1 and 2, GOV_INST and OTH_INST are negative and significant, suggesting that government and non-governmental blockholdings play a monitoring role and are not the direct effects that cause a higher cost of borrowing. Amongst the control variables, the coefficient estimates for AGE_FIRM ($\alpha = -2.322$, $p < 0.05$) are significant and negative. Thus, as expected, older companies, due to their reputation in the business community, seem to enjoy a lower cost of debt. Furthermore, LNSIZE and ROA are positive and significant, indicating that larger firms that are more profitable have a higher cost of debt. Our findings are consistent with those of the study by Wong et al. (2021), which finds that large Malaysian firms do not benefit from the possibility of lower default risks, better access to financing and economies of scale, as is the case in studies of listed firms in developed countries (Rajan and Zingales 1995). Thus, our findings seem contradictory to the trade-off theory that posits that profitable firms and firms with high tangible assets, all things being equal, should have lower bankruptcy risks and a lower cost of debt (Wong et al. 2021). As expected, leverage (CRED_POWER) increases the risk and the cost of debt, and growth firms (GROWTH) have a lower cost of debt after controlling for the financial and operational risks. Our results also suggest that, for politically connected companies, winning CSR awards increases their cost of borrowing because the coefficient for POL_CON is positive and significant. These findings extend the studies by Bliss et al. (2011) and Chaney et al. (2011) by showing that, for politically connected firms, it is not the governmental blockholding that is harmful but the overinvestment in national causes, such as those promoting CSR activities that are feared by creditors, including state-controlled banks. These findings lend credibility to these firm characteristics and our model with explanatory variables that significantly explain Malaysia PLCs' cost of borrowing.

## 5. Conclusions

This paper investigates the direct relationship between corporate social responsibility disclosures and winning CSR awards on the cost of debt. Our results indicate that, as the level of CSR disclosures becomes higher, the cost of debt becomes lower, which is consistent with the study by Richardson and Welker (2001). The authors revealed that disclosures directly affect the cost of capital. The findings by Anas et al. (2015) show that winning CSR awards is one of the factors influencing the level of CSR disclosures because companies want to build a good image and reputation in the market, which enables them to obtain cheaper financing or more favorable terms on their borrowing cost. To promote CSR culture amongst PLCs and to achieve Vision 2020, the government developed CSR awards with collaborations with external institutions, such as the media (for example, *The Star* newspaper) and Bursa Malaysia to create the *BRC*. When we tested for the interaction effect between CSR disclosures and CSR awards, our results show that, when companies win CSR awards, their cost of debt increases with negative economic consequences. This is

in line with the study by Goss and Roberts (2011), which indicates that overinvestment in CSR initiatives is a major concern of creditors. In addition, our results show that smaller boards are more effective in monitoring CSR reporting. Finally, the results show that politically connected companies have a higher borrowing cost, which is consistent with Bliss and Gul (2012).

Our findings contribute to the extent of the CSR literature by examining the influence of CSR disclosures and awards on the cost of debt in an emerging capital market. Our findings extend the studies by Dhaliwal et al. (2011) and Cuadrado-Ballesteros et al. (2016), which show that CSR disclosures are associated with a higher firm value and a lower cost of capital. We show that, as the CSR disclosures become greater, the cost of debt becomes lower, even though Malaysia does not have a formal CSR performance rating system. As such, investors and stakeholders find CSR disclosures and the associated rewards provided by the media as value relevant and rely on them to make decisions on capital allocation. When the moderation effects are considered, we find that, when companies win CSR awards, their cost of debt increases. The results suggest that Malaysia may have a negative perception of creditors for stakeholder orientation. This implication means that lenders do not rely on CSR disclosures, as they lack both third party assurances and quality (Cheng et al. 2017; Du and Vieira 2012). Therefore, Malaysian PLCs seem to use winning CSR awards as a legitimacy tool for survival and to have access to resources and finance. Thus, CSR disclosures do not play a complementary role in Malaysia (Amran and Susela Devi 2008). Prior studies by Kansal et al. (2014) and Sadou et al. (2017), focusing on emerging economies, show a positive relationship between CSR awards and CSR disclosures but do not investigate the relationship between these disclosures and the cost of debt. Sadou et al. (2017) showed that government ownership influences the level of CSR disclosures but lacks quality. These findings should interest regulators and policymakers to consider mandatory CSR disclosures.

However, this study has a limitation regarding testing the time lag factors influencing the cost of debt. We do not know how many years a particular factor should be lagged, and the results are not different when we lag them one period. However, generally, we believe factors such as ownership (both GOV_INST and OTH_INST) and reputation (i.e., from winning an award) do not change much from one period to the next. Moreover, loan officers evaluating investments likely do not pay much attention to short-term changes in these factors unless they are unusual and significant.

**Author Contributions:** S.D.S., S.M. and R.R. wrote this paper. S.D.S. conceived the idea, collected and analyzed the data and wrote the paper with S.M. and R.R. R.R. provided a methodical research design and formal analysis. S.M. made a final revision of the entire article together with reviewing and editing. All authors have read and agreed to the published version of the manuscript.

**Funding:** This research received no external funding.

**Institutional Review Board Statement:** Not applicable.

**Informed Consent Statement:** Not applicable.

**Data Availability Statement:** Data sharing does not apply to this article.

**Conflicts of Interest:** The authors declare no conflict of interest.

### Appendix A

The disclosure index acts as a proxy to measure the level of transparency and accountability. The derivation of the index is developed from the questionnaire prepared by StarBiz-ICR Malaysia. The Disclosure Index includes the following:

Corporate Social Responsibility, which is sub-sectioned as:

(1)   Community activities
(2)   Workplace: employment conditions
(3)   Environment

(4)   Reporting and standards

**Scoring Methodology**

**Industry/Sector**

One point is awarded if a company is in a sector which is not environmentally sensitive whilst two points are awarded if it is otherwise.

**Corporate Social Responsibility Strategy**

One point is scored if the CSR objective(s) is stated as part of the Mission/Vision statement of the company in the MD&A. One further point is awarded if the number of words exceeds the median number of words. Another additional point is given if the goal/strategy is quantified. If future actions by management are discussed, another point is given, and another one point is given if immediate actions are discussed. Thus, a company can receive a maximum of five points.

**Corporate Social Responsibility**

CSR is analyzed under the headings of community, workplace and environment.

**Community** consists of the structure of conducting community activities and the type of resources provided. Examples include encouraging employees to take part in community activities, such as planting trees and cleaning neighborhoods, and participating in the School Adoption Scheme. If disclosure is mediocre, one point is given; two points are given if it is disclosed in narration; and three points are given if it is quantified and in the narration form.

**Workplace** encompasses employment conditions, diversity, health and safety, and consultation (recommended by Bursa Malaysia's CSR Framework), with zero points for non-disclosure and one point if it is disclosed.

**Environment** includes climate change, energy, waste management, biodiversity and endangered wildlife. Zero points are given for non-disclosure, and one point is given if it /is disclosed.

Reporting and Standards

This section considers the type of sustainability reporting of companies in Malaysia. If a company observes a Code of Best Practice, one point is awarded. If it adapts an international standard, two points are given. If benchmarks are disclosed and compared to actual performance, three points are awarded.

**Appendix B**

| Variable Measurements | |
|---|---|
| Variable | Descriptions |
| Dependent | |
| Cost of Debt (COD_INT) | The interest expense of the firm divided by its average short-term and long-term data during the year. |
| CSR disclosures (CSR_INDEX) | The CSR_INDEX is obtained by calculating the total scores given to each firm under the StarBiz awards criteria. Thereafter, the median adjusted score for each firm within the industry is added to the number of words describing the CSR policy of the firm (items shown in Appendix A). |
| Board Governance | |
| Board Size (BO_SIZE) | Number of directors on the board. |
| Audit Committee Independence (AC_IND) | Percentage of independent directors on the audit committee. |
| Ownership Structure | |
| Government Institutional Investors (GOV_INST) | Percentage of shares of a company held by government-related institutional investors, directly or indirectly (EPF, LTAT, PNB, LTH and SOCSO). |
| Other Institutional Investors (OTH_INST) | Percentage of shares held by all other institutional investors (excluding EPF, LTAT, PNB, LTH and SOCSO) holding at least 5 percent of outstanding shares. |

|  | Variable Measurements |
| --- | --- |
| Political Connection (POL_CON) | An indicator variable that equals 1 if the company is politically connected and 0 otherwise. |
|  | Variable Measurements Descriptions |
| Variable | |
| CSR Awards (CSR_AWARDS) | An indicator variable that equals 1 if the firm was awarded a CSR award during that year and 0 otherwise. |
| Control | |
| Firm Size (FIRM_SIZE) | The log of the book value of total assets at the end of the year. |
| Age of Firm (FIRM_AGE) | Years since the firm began its business activity. |
| Return on Assets (ROA) | Profit after tax, divided by total assets. |
| Growth (GROWTH) | Tobin's Q as a proxy. Calculated as the market value of equity plus the book value of long-term debts and current liability, scaled by the book value of total assets. |
| Creditors' Power/Leverage (CRED_POWER) | Long-term debts divided by total assets. |

## Notes

[1]   The 10th Malaysia Plan identifies 12 key economic sectors that have the potential to achieve the status of a high-income nation. The sectors identified are oil and gas, palm oil and related products, financial services, tourism, education services, communication, information technology, wholesale and retailing, electric and electronics, business services, private health care, agriculture and the development of Greater Kuala Lumpur. The plan further states that this growth should be economically and environmentally sustainable, meeting the present needs without compromising those of future generations. (https://pmo.gov.my/dokumenattached/RMK/RMK10_Eds.pdf, assessed on 9 June 2022).

[2]   Though Malaysia is in the process of harmonizing its accounting standards with the International Accounting Standards, its current accounting standards, FRS 137 and FRS 101, do not consider social and environmental information in the Management Analysis and Discussion section of their annual reports. However, companies have been encouraged to provide information such as contingent liabilities and assets to aid investors in their decision making. Malaysia's environmental and social reporting practices are presently seen as a Code of Best Practice, and there are no specific accounting standards to address CSR reporting practices.

[3]   Under the legitimacy theory, a social contract exists in implicit terms between business and society. Friedman (1970) argued that a firm's primary purpose is to maximize shareholder wealth. A paradigm shift in the 1980s saw a change in the role of firms to that of meeting the needs of all stakeholders (Freeman 2010).

[4]   We believe that even private banks value PLCs with good CSR reputation, because, for example, the CEO of a leading private bank was the brother of the prime minister during the period of our study.

[5]   The largest banks in Malaysia are state owned, and most Malaysian PLCs have significant relations with them.

[6]   The Edge Billion Ringgit Club (*BRC*) members comprise about 19% of the number of listed companies in Bursa Malaysia as of 2014. Awards to companies are not only based on their financial performance, but due recognition is also given to companies showing exemplary leadership in building their businesses and creating value for their stakeholders.

[7]   The five largest public institutional investors in Malaysia are the Employees' Provident Fund (EPF), Lembaga Tabung Angkatan Tentera (LTAT), an investment fund (Permodalan Nasional Berhad (PNB)), a pilgrim fund (Lembaga Tabung Haji (LTH)) and the National Social Security Organisation of Malaysia (SOCSO). The Ministry of Finance defines these five institutional investors as the primary government-related institutional owners.

[8]   A possible reason for the average cost of debt of 5.2% is due to the reduced cost of borrowing from sources of funding such as grants, feed-in-tariff mechanisms and the Green Technology Financing Scheme (GTFS) offered by commercial banks, Islamic banks, Pembangunan, SME Bank, Agrobank, Bank Rakyat, EXIM Bank and Bank Simpanan Nasional.

[9]   We used the Janis–Fadner Coefficient to directly capture the effect of media influence on CSR_INDEX and COD_INT, but unfortunately, it was not significant.

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
