# Peer review of "CSR Disclosures, CSR Awards and Corporate Governance as Determinants of the Cost of Debt: Evidence from Malaysia"

_ijfs, doi:10.3390/ijfs10040087_

Round 1

Author Response

Reviewer 1 did not provide detailed feedback about the weaknesses in our article. The reviewer generally noted that the following areas must be improved in responding to the questions “Does the introduction provide sufficient background…”, “Are all the cited references relevant to the research”, “Is the research design appropriate”, “Are the results clearly presented” and “Are the conclusions supported by the results.” Thus, we are not able to provide point-by-point responses to address specific points made by Reviewer 1.

We would like to affirm that we have completely revised the paper and greatly improved the introduction section, the description of the research design, the writing of the results section, and making sure the results support the conclusions. Further, we have provided point-by-point responses to address Reviewer 2’s comments below, which could be taken as evidence of the significant revisions we have made to the current version of our article.

Reviewer 2 Report

The article is interesting, and proposes good hypotheses for testing. The methodology chosen (panel data with fixed effects model) is appropriate for the data.

The paper can spend more time to address the issue if CSR disclosures or awards can also act as proxies for advertising or branding the PLCs, especially on national media. Can such media exposure logically make these PLCs more marketable, increased profits, and thereby also allow them to negotiate better debt costs?

The State Banks are identified as one of the money lending institutions. Since the Malaysian government is actively promoting CSR disclosures as per the article, are these State Banks also obliged to follow these government instructions to prefer such companies that disclose CSR activities? Are the companies aware of any such CSR-favoring directives from the government to the State Banks?

There are only brief mentions of what actually constitute CSR activities in Malaysia. The article pointed out that Malaysia does not have a formal CSR performance rating system – this make it even more important to describe to a reader what is considered as CSR activities in Malaysia. Perhaps an appendix to describe these activities, and their related awards, will be useful.

The definition of politically-connected PLCs is not fully given in the article. Does this mean ownership of the company by a politician? Does it mean that a politician sits on the board? Are these politicians elected officials in power, or members of the opposition? Does this include family members of politicians, etc.

The definitions of GOV_INST and OTH_INST are not clear. Each is separately defined as based on percentage of shares owned (above 5%) by the respective institutions. However, are there companies simultaneously owned by both government institutions and other institutions, each with more than 5% ownership? Clearing up this question will improve understanding of both definitions.

Is the factor of lag addressed in the article? Does it take time for CSR disclosures or awards, board size changes, political connections etc. to affect debt costs, or are the effects assumed to be immediate, i.e., happening in the same time period?

Minor issues:

Page 1: “These national policies enabled Malaysia to attain high–income and developed nation status by 2020” – this statement contradicted later statements in the article stating that Malaysia is a developing country.

Page 2: “…the concept of a ’Green Economy,…” – should the apostrophe in front of “Green” be removed? (Unless it is a special term used for specific programs).

Page 13: Appendix A should not be inserted into the text – it will be sufficient to just refer to Appendix A in the text.

Page 15: Appendix B looks more like a Table, and perhaps should be converted to be one. If rewritten as a Table, it will be fine to insert it into the text at that location.

Page 21: “We show that the greater the CSR disclosures, the lower the debt cost. Even though Malaysia does not have a formal CSR performance rating system.” – these two statements look like they should be joined as one sentence. Alternatively, the second statement need to be rephrased.

Page 32: Table 2 – Descriptive Statistics – These statistics should also include the minimum and maximum numbers. Showing a range will should better show the spread of the data.

Round 2

Reviewer 1 Report

My main comments in the previous report were not taken into consideration in the current revision.
